# Application of the Controlled Source Radiomagnetotellurics (CSRMT) in the Study of Rocks Overlying Kimberlite Pipes in Yakutia/Siberia

Alexander K. Saraev [1,*], Arseny A. Shlykov [1] and Buelent Tezkan [2]

1   Institute of Earth Sciences, St. Petersburg State University, 199034 St. Petersburg, Russia; a.shlykov@spbu.ru
2   Institute of Geophysics and Meteorology, University of Cologne, 50923 Cologne, Germany; tezkan@geo.uni-koeln.de
*   Correspondence: a.saraev@spbu.ru; Tel.: +7-921-799-43-16

**Abstract:** The task of searching for kimberlite pipes in covered areas of the Yakutia kimberlite province is very difficult due to the significant heterogeneity of the rocks overlying kimberlite pipes. The overlying strata of terrigenous sediments contain rocks of the trap complex (dolerite sills, tuff bodies). We consider the results of the controlled source radiomagnetotelluric (CSRMT) soundings in Yakutia/Siberia. Due to the great thickness of the overlying rocks (near 100 m) and the relatively small horizontal sizes of kimberlite pipes (80–200 m), they cannot confidently be detected directly. An additional difficulty in identifying pipe anomalies is the presence of a layer of conductive carbonaceous siltstones in the overlying strata. Therefore, the main aim of the CSRMT surveys was the study of overlying rocks and the search for indirect indications of the presence of pipes. Possibilities to study the structure of dolerite sills located within overlying sediments and to map the top edge of hosting carbonate rocks are demonstrated using the CSRMT data. The pinching out of dolerite sills above pipes («windows in traps») and the lowering of the top edge of hosting rocks at pipes can be considered as indirect indications of the presence of pipes.

**Keywords:** kimberlite; overlying rock; controlled source radiomagnetotelluric; indirect indication





## 1. Introduction

The exploration for kimberlite pipes in Yakutia/Siberia in areas overlain with thick terrigenous sediments and traps is challenging with geophysical methods. Rocks of the trap complex of the intrusive phase (dolerites) and the effusive phase (tuffs) are characterized by significant variability in shape and physical properties. In areas with a low thickness of overlying sediments (up to 30 m), ground-based and airborne geophysical methods (magnetic and electrical) solve the problem of the exploration for kimberlites quite reliably. However, the effectiveness of geophysical methods is significantly reduced in areas with traps and a large thickness of sediments.

Kimberlites have resistivities of about 10–400 Ωm, dolerites—1000–2000 Ωm, tuffs—15–100 Ωm, hosting carbonate rocks—150–7000 Ωm, and overlying sediments—20–400 Ωm [1]. Therefore, kimberlite pipes usually have a low resistivity contrast compared to the overlying sediments and tuffs.

In areas covered with terrigenous sediments and with rocks of the trap complex (dolerites, tuffs), the anomalies of pipes are significantly less than the field variations associated with the heterogeneous structure, as well as changes in the properties of overlying strata, including traps. Often the horizontal sizes of the pipes are small, comparable to the thickness of the overlying rocks, and they create additional difficulties in identifying the pipes. However, the indirect indications of the presence of kimberlite pipes in the overlying and hosting rocks (presence of faults, structural features of rocks and changes of their properties) are often significant. Therefore, the detection and exploration of the

structural features and properties of surrounding rocks can be useful for the search of pipes. In order to identify anomalies of pipes according to the data of different ground-based and airborne geophysical methods, it is also necessary to take into account the structural features and properties of the overburden. The obtained information is also useful for their data interpretation.

Searches for kimberlite pipes in the covered areas of Yakutia are carried out by geological [2–5] and by geophysical [1,6–9] methods. The studies of structures of kimberlite fields and areas at pipes are considered as important prospecting tasks [10–13]. In these areas, the wild-cat drilling of prospecting wells with depths up to 150 m is carried on grids of 500 × 500 m and 250 × 250 m. In addition to the detection of pipes by drillings, the drilling data are used for studying, by mineralogical methods, the buried aureoles associated with diamond satellite minerals. The core of prospecting wells for this purpose is usually taken from the weathering interval of the hosting carbonate rocks. At the same time, satellite minerals can be located not only in the weathered zone at the contact of hosting rocks and overlying sediments, but also at different levels in the overlying strata due to the secondary redeposition of weathered rocks. Difficulties in the study of redeposited weathered rocks are associated with the significant heterogeneity of the overlying strata with a frequent pinching out of layers and a replacement of their facies. The information about the structure of the overburden obtained from the results of ground-based geophysical methods can be used to determine the depths of coring in the sediments. In addition, this information is useful for the paleo reconstructions when interpreting data of mineralogical methods.

This paper presents the results of the first application of the controlled source radio-magnetotelluric (CSRMT) method to study sedimentary rocks and traps overlying kimberlite pipes in the Alakit-Markha kimberlite field of Yakutia/Siberia in Russia (Figure 1).

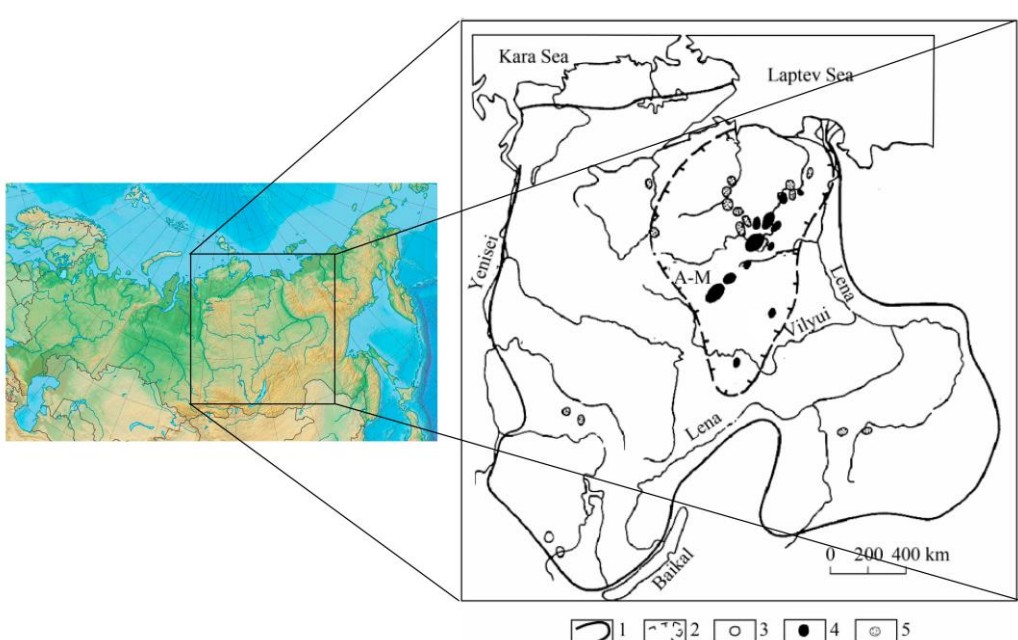

**Figure 1.** Map of Russia and the distribution of kimberlite rocks on the Siberia platform [2]. 1—Siberia platform boundary, 2—Yakutia kimberlite province boundary, 3–5—kimberlite fields of lower (3) and middle (4) Proterozoic and Mesozoic (5) age. A–M—Alakit-Markha kimberlite field (results of surveys in this field are considered in this paper).

## 2. Structure of Dolerite Sills in Kimberlite Prospecting Areas

In covered areas of the Yakutia kimberlite province, the overlying terrigenous sediments (sands, sandstones, siltstones, clays) contain intrusive dolerite sills and effusive tuffs. In the central parts of volcanic structures, the sills have a flat form and a large thickness (60–80 m and more). They usually penetrate between the hosting carbonate rocks and the

overlying sediments. Sometimes, these sills cut off the upper parts of pipes. In the side parts of volcanic structures, the sills have a smaller thickness (up to 30–50 m) and usually penetrate into the higher horizons of overlying sediments. Their shape becomes more complicated, often with abrupt changes in thickness and a complete pinching out [6]. The shape of sills is a very sensitive indicator of the structure of terrigenous sediments into which they penetrate. Changes in the shape of sills are especially noticeable near kimberlite pipes. The sills are often pinched out completely above the pipes. These structural features of sills—"windows in traps"—are considered as one of the indirect indications of the presence of kimberlite pipes [5,13].

Figure 2 shows the geological section through the kimberlite pipe Vostok in the Alakit-Markha field of Yakutia (see Figure 1), derived from the drilling data. The pipe is overlain with terrigenous sediments and dolerites. As seen from Figure 2a,b, the dolerite sill is pinched out above the pipe forming the "window in traps". A characteristic feature of the near-pipe space is the lowering of the top edge of hosting carbonate rocks. After the intrusion of the kimberlite pipe, the most favorable conditions for the denudation and erosion of hosting rocks were in the immediate vicinity of the pipe due to the presence of a zone of increased fracturing. As a result, an erosional depression was formed which was filled at the subsequent sedimentation. Like the "windows in traps", the depressions in the roof of hosting rocks can be considered as indirect indications of the presence of the pipe.

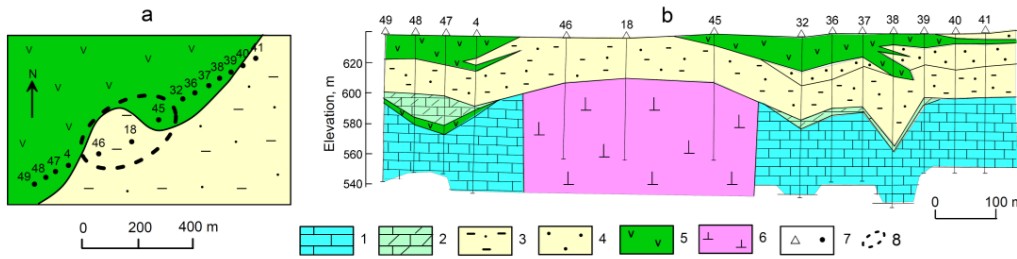

**Figure 2.** Features of the structure of dolerite sills and the top edge of hosting carbonate rocks near the kimberlite pipe Vostok in the Alakit-Markha field of Yakuta. (**a**)—plan, the location of wells, (**b**)—section along wells 49–41. 1—carbonate rocks, 2—fractured carbonate rocks, 3—sandstones and siltstones, 4—sands, 5—dolerites, 6—kimberlites, 7—well and its number, 8—contour of the pipe.

The presence of altered zones in the hosting carbonate rocks around the pipe is confirmed by the drilling data. To the left of the pipe, at the wells 48 and 47 in Figure 2b, an altered zone is identified near the top edge of the hosting rocks. A thin layer of dolerites near the top of the hosting strata was intruded along the contact between dense and altered carbonate rocks. To the right of the pipe, there is a noticeable decrease in the top edge of hosting rocks due to the karst at the wells 32 and 38, as determined from drilling data. In the overlying terrigenous sediments near the pipe there is a subsidence structure connected with the lowering of the top edge of hosting rocks. The dolerite sill was intruded into the overlain sediments. It has a similar shape as the subsidence structure and is characterized by a complicated form with a splitting into several thin sublayers. It should be noted that the geometry of dolerite sills is a very sensitive indicator of the structural features of rocks into which these sills are intruded. By studying the structure of sills, it is possible to obtain data about the structures of sedimentary rocks.

Changes in the form of kimberlite pipes and surrounding rocks during the geological history can be illustrated by schematic figures. After the explosive phase of a pipe's penetration, a pipe has a cylindrical or conical channel and a bell in the upper part of the body (Figure 3a). Explosive formations are presented on the surface of the earth. Near the pipe's body, zones of increased fracturing are formed. Further, as a result of the processes of denudation and erosion of the hosting carbonate rocks, the upper part of the pipe was cut off. At the same time, the erosion depression was formed in the zone of increased fracturing due to hydrothermal processes and the destruction of carbonate rocks (Figure 3b). In the

course of subsequent sedimentation, the layers of overlying rocks were accumulated, and the subsidence structure in zones of the increased fracturing was formed. The shape of the top edge of the hosting carbonate rocks was inherited in the overlying sedimentary strata (Figure 3c).

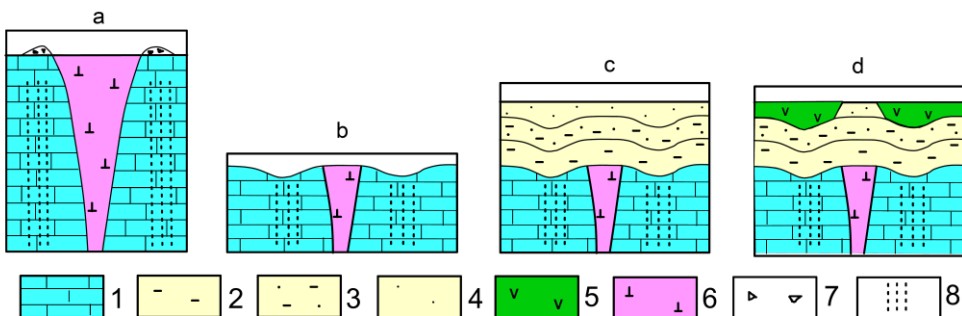

**Figure 3.** Scheme of evolution of kimberlite pipes and surrounding rocks. (**a**)—pipe after the explosion, (**b**)—pipe after the denudation and erosion, (**c**)—pipe after the sediments' accumulation, (**d**)—pipe after the intrusion of dolerite sill. 1—carbonate rocks, 2—carbonaceous siltstones, 3—sandstones and siltstones, 4—sands, 5—dolerites, 6—kimberlites, 7—explosive formations, 8—fractures zones.

Then, at the stage of trap magmatism in the side parts of volcanic structures, a dolerite sill penetrates into the overlying sediments. Its intrusion essentially depends on the structure of sediments. The dolerite magma flowed around the positive structures in the section changing the shape of sill up to its complete pinching out.

The formation of the "window in traps" over a local raising in sedimentary layers above the pipe is illustrated by Figure 3d. The considered scheme of the formation of "windows in traps" above the pipes and depressions in the roof of carbonate rocks near the pipes is also consistent with the publications of [2,5,13].

## 3. Principles of the CSRMT

The method CSRMT is based on measurements of the electromagnetic fields of a controlled source operating usually in the frequency range of 1 kHz to 1000 kHz (in this experiment in addition we transmitted and measured signals on the frequency 0.5 kHz but only highest subharmonics were used because of the frequency range of the receiver). Using data of measurements of the horizontal and mutually perpendicular components of the electrical and magnetic fields, the impedance magnitude (apparent resistivity) and the impedance phase are calculated. The tipper parameters are determined from the measurements of the three components of the magnetic field. Sounding curves are derived from the frequency dependence of the apparent resistivity and of the impedance phase. Their joint 2D inversion allow us to derive the resistivity distribution beneath the investigated profile depending on the penetration depth of the used frequencies.

In the CSRMT method, a horizontal magnetic dipole (vertical loop) [14] or a horizontal electrical dipole (finite-length wire) [15,16] can be used as a source. The use of the horizontal magnetic dipole has several advantages (e.g., the compactness of installation, the possibility of simple implementation for tensor measurements). However, the disadvantages of this source (limited operating frequency range 1–12 kHz, small long-range action, no more than 600–800 m, the impossibility of using subharmonics of a fundamental frequency, and the necessity to adjust the source to resonance at each frequency) can limit its application in field surveys.

More promising is the use of the horizontal electrical dipole as a source in the CSRMT method. Measurements up to a distance of 3–5 km are possible using this source. Four or five base frequencies and their subharmonics were used to cover the frequency range from 1 kHz to 1 MHz of the electromagnetic field in the survey area. We use ungrounded (capacitive) electrical antennas to measure the electrical field, thus making it possible

to carry out field surveys in winter under unfavorable conditions for the grounding of electrodes (frozen ground, snow, ice). The magnetic field components are measured using induction coil magnetometers.

In this case study, we applied the scalar variant of CSRMT measurements. We used a single grounded wire directed in the northwestern direction and measured the horizontal electrical field $E_X$ along the wire and horizontal magnetic field $H_Y$ across the wire. Therefore, only one component of the impedance was calculated $Z_{XY} = E_X / H_Y$ along the direction of profiles. Taking into account the northeastern direction of the main fault zone in this area [2], we can conclude that the impedance $Z_{XY}$ corresponds to the case of H-polarization. Vertical magnetic component was also measured, but because of the large Tx-Rx offset the amplitude of $H_Z$ was too small to be confidently used in data analysis and inversion.

### 4. Results of the CSRMT in the Kimberlite Area Covered with Traps

Kimberlite pipe TSNIGRI covered with terrigenous sediments and traps is located in the Alakit-Markha kimberlite field (see Figure 1). The thickness of the overburden is about 100 m (Figure 4). They include terrigenous sediments of the Carboniferous and Permian ages (sandstones, siltstones) and rocks of the trap complex (dolerites and tuffs). Hosting rocks are presented by limestones, marls and dolomites of the Lower Paleozoic age. In Figure 4a the layout of the pipe and positions of wells are shown. Figure 4b illustrates the section along these wells.

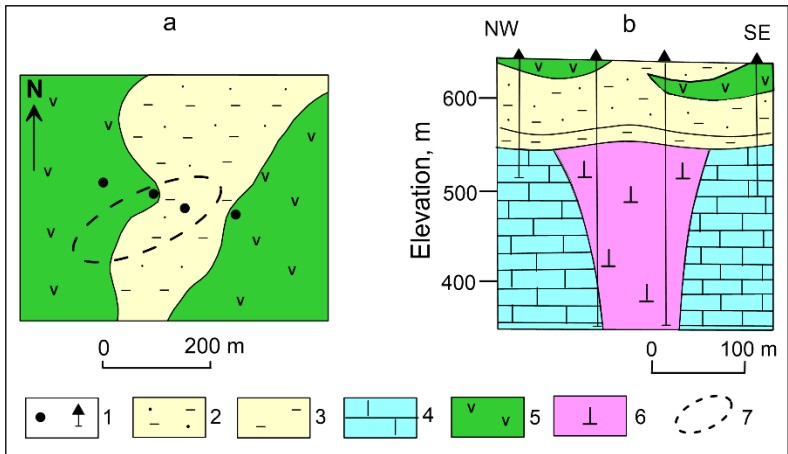

**Figure 4.** Layout of the pipe TSNIGRI (**a**) and the section through the pipe (**b**). 1—wells, 2—sandstones and siltstones, 3—carbonaceous siltstones, 4—carbonate rocks, 5—dolerites, 6—kimberlites, 7—contour of the pipe.

The use of the standard radiomagnetotelluric (RMT) sounding method in the frequency range 10–1000 kHz [17] is ineffective due to the small number of radio transmitters in this region. In such a remote area, the broadcasting VLF transmitters are only available in the frequency range of 10–30 kHz, which is not enough for soundings. In addition, due to the high thickness of the overlying rocks, it was necessary to measure electromagnetic fields at low frequencies to reach the target depth of investigation. This is possible using the controlled source, which has the extended frequency range of 1–1000 kHz.

When carrying out the field work in the survey area by the CSRMT method, a 1000 m long wire grounded at the ends was used as a source. It was installed in a northwestern direction and located at a distance of about 2700–3000 m from the site (Figure 5). The soundings were performed in the broadside area of the source in a scalar variant along the profiles of the northwestern direction, crossing the pipe along its short axis and dolerites (see Figure 4).

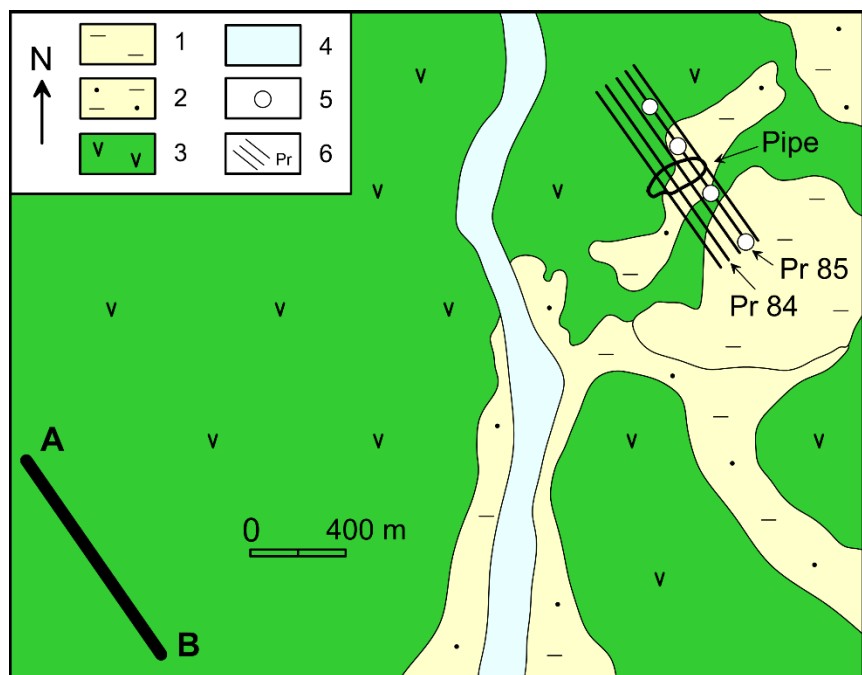

**Figure 5.** Scheme of location of the source AB and the CSRMT profiles on the kimberlite pipe TSNIGRI. 1—carbonaceous siltstones, 2—sandstones and siltstones, 3—dolerites, 4—river and Quaternary sediments, 5—CSRMT stations on the profile 85, from SE to NW: St.00, St.30, St.50 and St75, 6—CSRMT profiles.

The main aim of the survey near the pipe TSNIGRI was the study of the structure of dolerite sills and the relief of the top edge of the hosting carbonate rocks. The distance between profiles and sounding stations was 50 m (95 stations in total). Each profile was 1000 m long. The receiving electrical lines were directed along the profile, and magnetic antennas were perpendicular to the profile. The base operating frequencies of the transmitter were set at 0.5, 1.1, 10.1, 90, and 115 kHz and the signals were measured at these frequencies and at their odd subharmonics. For the 0.5 kHz we used the subharmonics only (1.5, 2.5, . . . , 9.5 kHz) because of the frequency range of the receiver.

Examples of the sounding curves for several stations located on profile 85 are shown in Figure 6. This figure illustrates the main types of sounding curves in this area. Measured data are indicated by dots. For the validation of the far-field approximation, we performed 1D controlled source inversion accounting the transition zone (or source) effect using the code described in [18]. The 1D inversion code is based on a full solution of a finite length and arbitrary shaped wire in the horizontally layered media. For the far field validation, we used simple layered inversion with Levenberg-Marquardt regularization schema [19]. For selecting the optimal damping parameter, we used an empirical approach. For each iteration we "tested" 20 damping parameters loglinearly distributed between $10^5$ and $10^{-3}$ and selected the value, which corresponds to minimal misfit. For more precise estimation of the optimal damping, we applied spline approximation of the damping-versus-misfit functional around the preselected above value. The fitted synthetic CSRMT responses are indicated by solid lines. Dashed lines show the synthetic plane wave response for the model after 1D inversion, including the transition zone effect. In the far-field zone, the controlled source and plane wave responses are the same. They differ in the transition zone [20]. We can see that the transition zone effect is visible on the first (lowest) frequency of the apparent resistivity curves and one or two of the lowest frequencies on the phase curves. For example, on the station 75, where we can see very strong response from the resistive layer, the far-field zone conditions are valid for all frequencies. Using such simple analysis, we can exclude the transition zone effect for further 2D plane wave inversion.

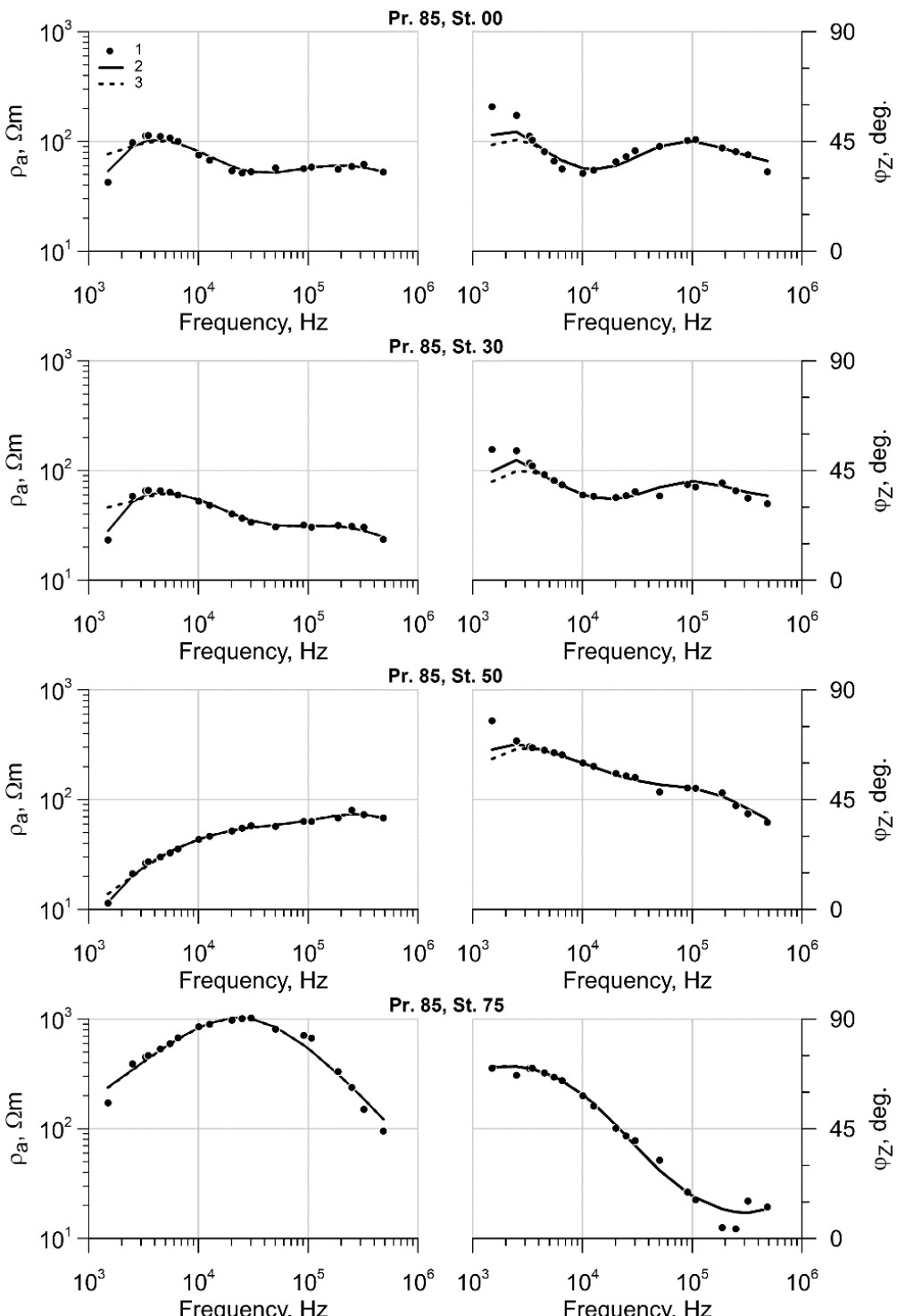

**Figure 6.** Examples of CSRMT apparent resistivity and phase curves. 1—measured data, 2—synthetic data after 1D controlled source inversion considering the transition zone, 3—synthetic plane wave responses for the model after controlled source inversion. The positions of selected stations are marked in Figure 5.

Figure 7 shows pseudo-sections of the apparent resistivity and the impedance phase for five profiles. The boundaries of the pipe are schematically shown in the pseudo-sections. The analysis of pseudo-sections shows that they have a similar view on adjacent profiles. The similarity of pseudo-sections of the apparent resistivity along different profiles indicates

the relatively small effect of static shifts on the sounding curves for the high-frequency CSRMT method.

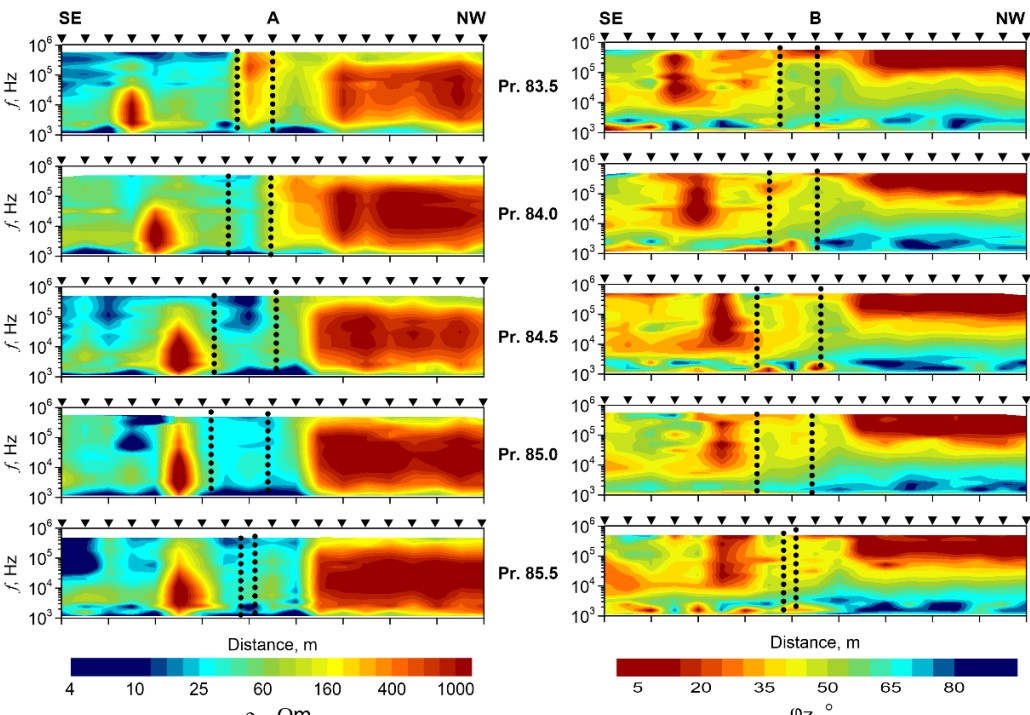

**Figure 7.** Pseudo-sections of the apparent resistivity (**A**) and impedance phase (**B**) along the profiles at the kimberlite pipe TSNIGRI. Black dashed lines—contour of the pipe, black triangles—sounding stations.

As expected, the effect of the pipe cannot be recognized on the pseudo-sections of the CSRMT data because of the large thickness of the overlaying sediments and low resistivity contrast with hosting rocks. However, the study of the overburden was considered as the main research focus for the survey area and the pseudo-sections definitely illustrate the variation of resistivity along the profiles. In addition, a complicating factor for identifying the pipe is the presence of a layer of highly conductive carbonaceous siltstones. According to the results of induction logging (IL), their resistivity is 20–60 $\Omega$m [9]. They have a thickness of about 20–30 m and lie above the top edge of hosting carbonate rocks. The resistivity of the hosting carbonate rocks was also estimated using data of IL as 500–2000 $\Omega$m, in spite of the limited range of measurements in IL (confident measurements are possible for a resistivity of no more than 500 $\Omega$m). An estimation of the resistivity of the dolerites is unfortunately impossible from the IL data [1].

Two-dimensional plane wave inversion was performed using the commercial software ZondMT2D (zond-geo.com). Examples of the pseudo-sections of the initial and reached misfits for the profile 85 are presented in Figure 8. Relatively big misfits for apparent resistivity at the lowest frequency indicate the transition zone effect. These frequencies were not used during the inversion. Big phase misfits at high frequencies in the NW part of profile were caused by small phase values, at only 2–5°. Therefore, the moderate absolute difference between the measured and predicted phases normalized by the small measured value will lead to a relatively big error. The averaged RMS misfit for apparent resistivity after 20–40 iterations was 6–8% and for the phase was 4–6%.

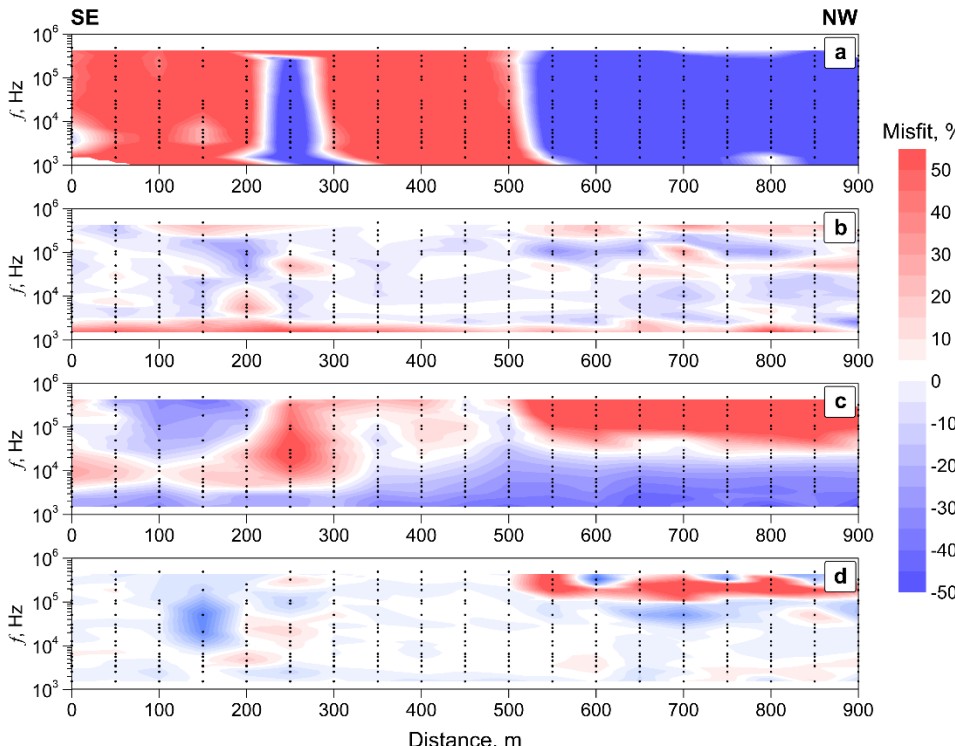

**Figure 8.** Pseudo-sections of the initial misfits for the start model (100 Ωm half space) reached misfits after converged inversion along the profile 85 at the kimberlite pipe TSNIGRI. (**a**)—initial misfits for apparent resistivity, (**b**)—reached misfit for apparent resistivity, (**c**)—initial misfit for impedance phase, (**d**)—reached misfit for impedance phase. Not fitted apparent resistivity at the lowest frequency corresponds to the transition zone (did not used in the inversion). Black dots indicate data points.

The resulting 2D resistivity sections are characterized by the smooth behavior of the boundaries and a good correlation from profile to profile (Figure 9). The top edge of the hosting carbonate rocks (A) just below conductive carbonaceous siltstones is confidently mapped. The most contrasting and confidently distinguished is the conductive layer of carbonaceous siltstones (C), with resistivity of 10–50 Ωm. Dolerite sills (D) are marked by high resistivity values. Dolerites are pinching out directly above the pipe forming the "window in traps". The resistivity of SE layers of dolerites is a bit less than that of NW layers. The possible reason for this is the weathering of the dolerites in the SE part of the profiles. The layer of sandstones and siltstones (B) above the carbonaceous siltstones is characterized by resistivity values of 100–300 Ωm. There is also a lowering of the top edge of the hosting rocks near the pipe. As noted above, these features of the structure of rocks of the trap complex and the structure of the top edge of hosting rocks can be considered as indirect indications of the presence of kimberlite pipes.

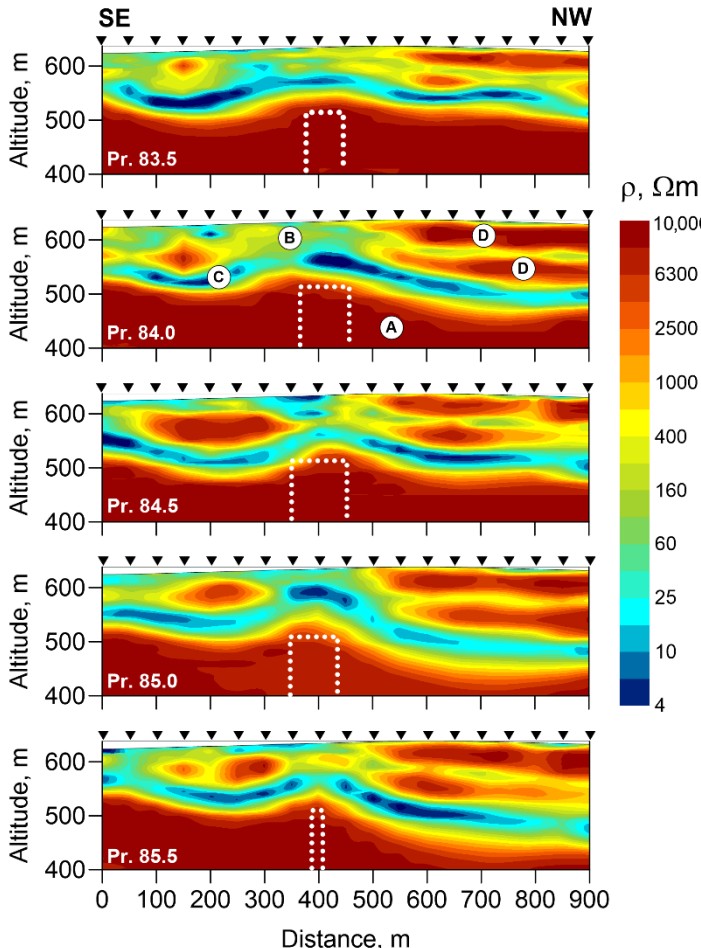

**Figure 9.** Resistivity sections along the profiles at the kimberlite pipe TSNIGRI. White dashed lines—contours of the pipe, black triangles—sounding stations. Blocks in resistivity sections: A—hosting carbonate rocks, B—overlying sediments, C—carbonaceous siltstones, D—dolerites.

## 5. Results of the CSRMT in the Kimberlite Area Covered with Terrigenous Sediments

Kimberlite pipe Baitakhskaya is located in the Alakit-Markha kimberlite field (see Figure 1). It is covered with terrigenous sediments with a thickness of about 60–70 m. The overburden contains mainly sandstones and siltstones of the Carboniferous and Permian ages. The hosting rocks are presented by limestones, marls, and dolomites of the Lower Paleozoic age. Figure 10 shows the geological section through the Baitakhskaya pipe.

The distance between the profiles of 1150 m length and the stations at this site was 50 m, the azimuth of profiles was 0°. The controlled source (1000 m length wire) was located at a distance of 2 km from the western boundary of this site (Figure 11). The direction of wire was selected as in the previous case for the pipe TSNIGRI approximately across the main fault zone in this area. Validation of the far-field zone conditions was performed here in a similar manner as described in previous section.

The main aim of the survey near the pipe Baitakhskaya was the study of the detail structure of overlying sediments. The soundings were carried out along the profiles in the scalar variant. The electric lines were directed along the profiles, the magnetic antennas were perpendicular to the profiles. In the course of the measurements, signals from the controlled source at the main frequencies of 0.5, 1.1, 10.1, and 107 kHz and their odd subharmonics were measured. In total, there were 69 CSRMT sounding stations.

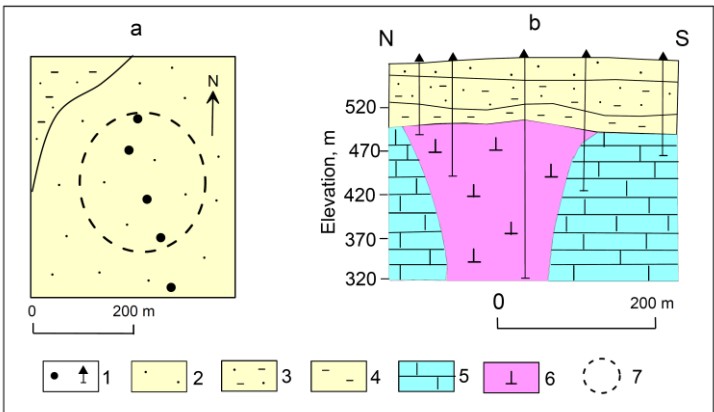

**Figure 10.** Layout of the pipe Baitakhskaya (**a**) and the section through the pipe (**b**). 1—wells, 2—sands, 3—sandstones and siltstones, 4—carbonaceous siltstones, 5—carbonate rocks, 6—kimberlites, 7—contour of the pipe.

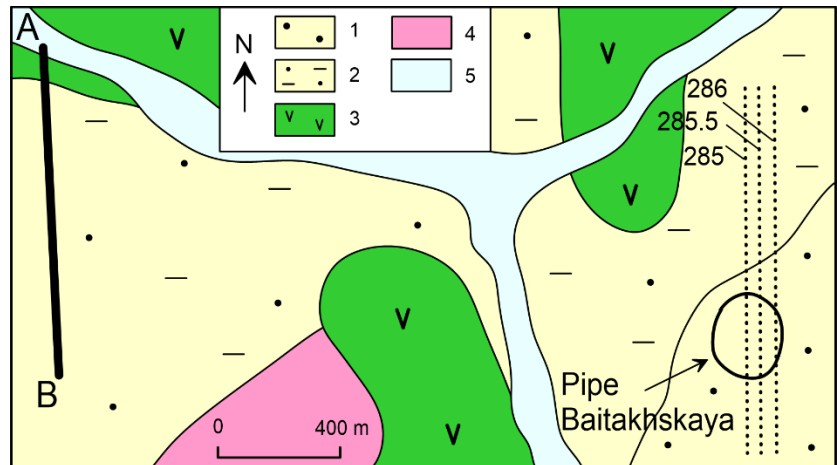

**Figure 11.** Scheme of location of the source AB and profiles at the kimberlite pipe Baitakhskaya. 1—sands, 2—sandstones and siltstones, 3—dolerites, 4—tuffs, 5—river and Quaternary sediments.

Figure 12 shows the resistivity sections along selected profiles crossing the pipes Baitakhskaya as results of the 2D inversion of the observed data using the ZondMT2D software (the RMS for apparent resistivity is, on average, 6%, and for the impedance phase is 4%). The sections also show schematically the borders of the pipe.

The resistivity sections in Figure 11 indicate a smooth behavior of the boundaries and their good correlation from profile to profile. The top edge of the kimberlite hosting rocks (A) just below conductive carbonaceous siltstones (C) is confidently mapped. The most contrasting and confidently distinguished is the conductive layer of carbonaceous siltstones. The features of the roof structure of hosting rocks, which were noted above when considering the results of work at the pipe TSNIGRI, are also observed near the pipe Baitakhskaya.

The obtained resistivity sections show that the CSRMT method makes it possible to study the detailed structure of overlying sediments, from the first meters to the roof of hosting carbonate rocks. The overlying strata (B) is divided into more conductive and less conductive sublayers of sandy and clayey sediments. At the same time, their pinching out along the section is clearly visible. The good repeatability of structural details of the overlying terrigenous strata on adjacent profiles confirms the reliability of the interpretation of CSRMT data. The data on the detailed subdivision and correlation of the overlying terrigenous sediments can be useful for the study of redeposited weathered crusts in the mineralogical methods of prospecting for kimberlites.

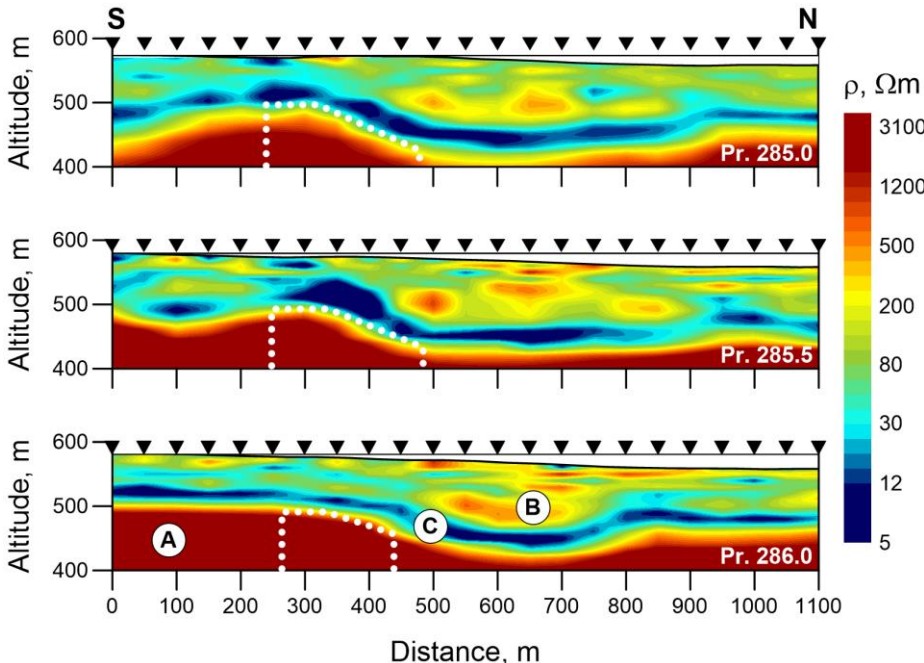

**Figure 12.** Resistivity sections along profiles 285, 285.5, 286. Black triangles—sounding stations, White dashed lines—contours of the kimberlite pipe Baitakhskaya. Blocks in resistivity sections: A—hosting carbonate rocks, B—overlying sediments, C—carbonaceous siltstones.

## 6. Conclusions

The structures of rocks overlaying kimberlite pipes in two areas located in the Alakit-Markha field of the Yakutia kimberlite province were studied by CSRMT field surveys. The CSRMT soundings were carried out in the far-field zone of the electrical dipole source (grounded at the ends wire) in the scalar variant in the frequency range of 1–1000 kHz. During the measurements, the signals of the fundamental frequencies and their subharmonics were used. The similarity of pseudo-sections of the apparent resistivity on different profiles indicates a relatively small effect of the static shift on the sounding curves for the high-frequency CSRMT method.

Based on the results of 2D data inversion, the capabilities of the CSRMT method are demonstrated for the detailed study of rocks overlaying the kimberlite pipes in two different areas of the Yakutia province. One area is overlain with terrigenous sediments and traps and the other with terrigenous sediments only. In both cases, we successfully mapped the conductive zone of the carbonaceous siltstones over the kimberlite hosting rocks, which provides us the possibility to trace the top edge of carbonate rocks and to identify depressions in the hosting rocks at the pipes. We obtained detailed resistivity sections of the overlain sediments with a good correlation between profiles. In the area overlain with terrigenous sediments and traps, we studied the structure of dolerite sills and identified the pinching out dolerite sills above the pipe, which form the "windows in traps".

The "windows in traps" and the depressions in the hosting rocks are characteristic features for the localization of kimberlite pipes. They are considered as indirect indications of the presence of pipes. The data on the structural features of the overlying sediments containing diamond satellite minerals can be used to interpret the results of mineralogical methods for the detection of kimberlites.

**Author Contributions:** Conceptualization, A.K.S.; methodology, A.K.S.; software, A.A.S.; validation, A.K.S., A.A.S., B.T.; formal analysis, B.T.; investigation, A.K.S., A.A.S., B.T.; resources, A.K.S., A.A.S.; data, A.K.S.; curation, B.T.; writing—original draft preparation, A.K.S.; writing review and editing, B.T.; visualization, A.K.S., A.A.S.; supervision, B.T.; project administration, A.K.S., B.T.; funding acquisition, A.K.S., B.T. All authors have read and agreed to the published version of the manuscript.

**Funding:** This work was funded by the Russian Science Foundation, project No 21-47-04401.

**Institutional Review Board Statement:** Not applicable.

**Informed Consent Statement:** Not applicable.

**Data Availability Statement:** The datasets generated and analyzed in the current study are available from the corresponding author on reasonable request.

**Acknowledgments:** The presented results were obtained with the support of the Russian Science Foundation, project No 21-47-04401, the GEOMODEL of St. Petersburg State University and the German Science Foundation, (project TE 170/21-1).

**Conflicts of Interest:** The authors declare no conflict of interest.

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
