# Peer review of "Application of the Controlled Source Radiomagnetotellurics (CSRMT) in the Study of Rocks Overlying Kimberlite Pipes in Yakutia/Siberia"

_geosciences, doi:10.3390/geosciences12010034_

Round 1
Reviewer 1 Report
Comments on the manuscript Application of the Controlled Source Radiomagnetotellurics (CSRMT) in the Study of Rocks Overlying Kimberlite Pipes in 3 Yakutia/Siberia by Saraev et al., Summary The manuscript describes the application of a near-surfcae controlled source EM experiment to characterize the overburden of kimberlit pipes in Yakutia, Siberia. The problem treated here is that kimberlite pipes cannot be imaged directly, but that dolorite layers associated with the kimberlite intrusion appear to show a distinct gap (a window) above the pipe head. This window is shown to be discernable using the applied exploration method. In this way, the research has found a means to indirectly infer the presence of kimberlite pipes. The manuscript is well and concisely written and the geological nackgorund, the method and the results are laid out very clearly. The authors have presented a nice piece of work, and I have only minor comments. Otherwise the manuscript can be almost accepted as it stands. Minor Comments:- line 92: “in the near-pipe space”, perhaps rephrase to “in the immediate vicinity of the pipe”
- line 93; “As a result, an erosional depression ...”
- Figure 3, panel d: Dolerites intruded into the interface between the carbonatic host rock and the overlying sediments are not depcited in this figure (cf. Fig. 2). Any reason for that?
- line 154, and following occurrences. The term subharmonics should rather be harmonics, shouldn't it?
- line 159. “... the whole frequency range ...”. Delete “whole”
- line 160: The frequency range given here is nit identical to the range given previously. Please make this consistent.
- lines 173-178. The paragraph is in parts repetitive to previous explanations; please revise.
- line 192: r>4d, make this an approximate measure, i.e. r≳ 4d
- Figure 6: It is quite strange that the phase exceeds 45 deg, while the app. res. increases over a broad band. Any explanation, why this is plausible? It would be nice to see a suite of sounding curves, not only one example. It would also be nice to see the predicted data from the inversion model. What is the dashed line? Are there noe error estimates available? Can you comment on the frequency distribution?
- Line 238, Inversion models. The resistivity of the carbonates are said to be fixed to 2000 Ohm-m. How was the topography of the carbonate layer determined (it is seemingly undulating in the 2D inversion images)?
- Section 5: the text is in parts repetitive to the previous section; try to minimize thus (e.g. remove the link to zond-geo.com etc). Instead, emphasize a bit more the motivation of the choice of these two sites, and what are there distinct characteristics.
Author Response
Dear reviewer,
Thank you for your comments and very useful suggestions for the improvement of the manuscript.
They were mainly taken into account and we made changes in figures and text.
Look please below our answers.
Yours sincerely,
Authors
Q: line 92: “in the near-pipe space”, perhaps rephrase to “in the immediate vicinity of the pipe”
A: Done
Q: line 93; “As a result, anerosional depression ...”
A: Done
Q: Figure 3, panel d: Dolerites intruded into the interface between the carbonatic host rock and the overlying sediments are not depcited in this figure (cf. Fig. 2). Any reason for that?
A: We observe a case when dolerites intrude into the overlying sediments. In this case they form “windows in traps”. When dolerites intruded between the carbonate hosting rocks and overlying sediments they usually have the bed form.
Q: line 154, and following occurrences. The term subharmonicsshould rather be harmonics, shouldn't it?
A: Actually they are the same but the “harmonics” is a bit more general but the “subharmonics” exactly mean that it is a sub harmonics of the main frequency.
Q: line 159. “... the whole frequency range ...”. Delete “whole”
A: DONE
Q: line 160: The frequency range given here is nit identical to the range given previously. Please make this consistent.
A: DONE
Q: lines 173-178. The paragraph is in parts repetitive to previous explanations; please revise.
A: DONE
Q: line 192: r>4d, make this an approximate measure, i.e. r≳ 4d
A: Sorry, we have no such symbol on our Word soft… This part is removed and replaced by other far-field validation procedure description.
Q: Figure 6: It is quite strange that the phase exceeds 45 deg, while the app. res. increases over a broad band. Any explanation, why this is plausible? It would be nice to see a suite of sounding curves, not only one example. It would also be nice to see the predicted data from the inversion model. What is the dashed line? Are there noe error estimates available? Can you comment on the frequency distribution?
A: Actually, in this figure apparent resistivity decreases toward the low frequency and it corresponds to increasing of the phase. It is usual situation. The frequency distribution is related to the selected base frequencies of the transmitter only. They are mentioned in the text (page 6).
Q: Line 238, Inversion models. The resistivity of the carbonates are said to be fixed to 2000 Ohm-m. How was the topography of the carbonate layer determined (it is seemingly undulating in the 2D inversion images)?
A: It is removed. It was a mistake…
Q: Section 5: the text is in parts repetitive to the previous section; try to minimize thus (e.g. remove the link to zond-geo.com etc). Instead, emphasize a bit more the motivation of the choice of these two sites, and what are there distinct characteristics.
A: Done.
PLease look at revised manuscript.

Reviewer 2 Report
The present version is not acceptable for publication. I could suggest a resubmission. The following are some specific comments:
- Better to provide a regional geological map for Figure 2.
- The section of “Principles of the CSRMT” is too short to present information needed in data processing, for example, how you correct the source effect? How you do the 2D inversion? For 2D inversion, which mode and which direction you used? And, why is that? Is there any geological evidence to support the selected direction? You might need to describe the inversion algorithm along with the data processing procedure.
- It’s also better to describe all the sounding curves, at least, the major types, with regard to site location, of sounding curves. Good thing is to arrange data presentation in a new section before Section 4. Just one curve is not enough to judge the inverted resistivity model. Sounding curves at all stations, when the authors decide to present, could be provided in the form of supplementary materials. Pseudo-section in, say, Figure 7 is, of course, one way to present the data, but should be given after the original curves.
- There is no way to evaluate the reliability of inverted models without fitting curves or calculated pseudo-sections.
- With regard to Figures 5 & 10, I don’t even see an informative map showing the locations of soundings. There is no spatial information overlaid with the reginal/geological maps for Figures 5 & 10. A good way is to combine them with Figs 4 & 9, together with more local geological information shown in those figs.
Author Response
Dear reviewer,
Thank you for your comments and very useful suggestions for the improvement of the manuscript.
They were mainly taken into account and we made changes in figures and text.
Look please below our answers.
Yours sincerely,
Authors
Q: Better to provide a regional geological map for Figure 2.
A: Regional map is too complicated and redundant. We added rocks to the plane view.
Q: The section of “Principles of the CSRMT” is too short to present information needed in data processing, for example, how you correct the source effect? How you do the 2D inversion? For 2D inversion, which mode and which direction you used? And, why is that? Is there any geological evidence to support the selected direction? You might need to describe the inversion algorithm along with the data processing procedure.
A: We extended this section but not so much because we used the scalar variant of the CSRMT method without any opportunity to have a choice of modes etc.
We have added the information about the northeastern direction of the main fault zone in this area, so we have installed the transmitter line in the northwestern direction. In this case the impedance ZXY corresponds to the case of H-polarization.
And this part is standard for several related publications we sited in this section. We used the commercial inversion software without detailed description of algorithm. Therefore we can’t really do this description here…
Q: It’s also better to describe all the sounding curves, at least, the major types, with regard to site location, of sounding curves. Good thing is to arrange data presentation in a new section before Section 4. Just one curve is not enough to judge the inverted resistivity model. Sounding curves at all stations, when the authors decide to present, could be provided in the form of supplementary materials. Pseudo-section in, say, Figure 7 is, of course, one way to present the data, but should be given after the original curves.
A: We added the main types of curves (4 types) with some additional synthetic data). Pseudo-sections are clearly show that we have few types of curves here and it is really redundant to present all curves.
Q: There is no way to evaluate the reliability of inverted models without fitting curves or calculated pseudo-sections.
A: Added.
With regard to Figures 5 & 10, I don’t even see an informative map showing the locations of soundings. There is no spatial information overlaid with the regional/geological maps for Figures 5 & 10. A good way is to combine them with Figs 4 & 9, together with more local geological information shown in those figs.
A: Done.
Reviewer 3 Report
It is difficult to see the conductive layer in the pseudo-sections as it seems to be a very subtle feature, yet its interpretation is crucial to the interpretation. Looking at the pseudo section, it is also very difficult to identify a priori where the kimberlite features are. These difficulties then made it hard to understand why the resistivity scale was chosen to go to 10-3 when the expected rocks were conductive but not much less than 10 ohm-m. Did the inversion require resistivities less than a few ohm m?. Does this make one suspect the data, calibration or other issues with the data?. It left me wondering how much confidence to this structure. The RMS is stated as 2.5% at the end – but how much improvement did the inversion bring to the modeling?
It would be nice to see a pseudo section of the starting and ending misfit in resistivity and phase to address these concerns (since the misfit should be random). As an aside I was wondering about issues of 2d structural assumptions implicit in the modeling and how failures of this assumption would be identified. These are general concerns that the authors could address to give more confidence to the reader. The authors note that the inverted resistivities are smoothly consistent along each profile and from profile to profile. This feature and the observed good fitting of the data gives the authors confidence in their interpretation. My concern is what makes it so difficult to see this structure in the raw pseudo-sections.
As the observed identification of the morphology of the carbonaceous layer and structure of the overlying sediments (or sills) is key, it would be helpful to have the authors give more support for their confidence in these impressive results. For the second case history, it would be helpful to show also the pseudo sections of the observed profiles. The goal would be to help the reader gain confidence in the resolving power of the method.
Author Response
Dear reviewer,
Thank you for your comments and very useful suggestions for the improvement of the manuscript.
They were mainly taken into account and we made changes in figures and text.
Look please below our answers.
Yours sincerely,
Authors
Q: It is difficult to see the conductive layer in the pseudo-sections as it seems to be a very subtle feature, yet its interpretation is crucial to the interpretation.
A: This feature of section is mostly visible on the phase pseudo-sections because it is the deep part of the section and for same frequency phase provides information about deeper horizons than apparent resistivity.
Q: Looking at the pseudo section, it is also very difficult to identify a priori where the kimberlite features are.
A: Sure. That’s why we used the approach connected with attempt to find indirect signs of pipes. If we can see the pipe in data we can use direct search without any troubles. But here another more complicated story.
Q: These difficulties then made it hard to understand why the resistivity scale was chosen to go to 10-3 when the expected rocks were conductive but not much less than 10 ohm-m. Did the inversion require resistivities less than a few ohm m?. Does this make one suspect the data, calibration or other issues with the data?. It left me wondering how much confidence to this structure.
A: Of cause, it was a mistake in figure… We revised sections.
Q: The RMS is stated as 2.5% at the end – but how much improvement did the inversion bring to the modeling? It would be nice to see a pseudo section of the starting and ending misfit in resistivity and phase to address these concerns (since the misfit should be random).
A: RMT values are also revised and an example of pseudo-sections of start and reached misfits is added.
Q: As an aside I was wondering about issues of 2d structural assumptions implicit in the modeling and how failures of this assumption would be identified. These are general concerns that the authors could address to give more confidence to the reader. The authors note that the inverted resistivities are smoothly consistent along each profile and from profile to profile. This feature and the observed good fitting of the data gives the authors confidence in their interpretation. My concern is what makes it so difficult to see this structure in the raw pseudo-sections.
A: Only pipe is “invisible” in data pseudo-sections. Other structural features like windows in traps, lowering of the top edge of hosting carbonate rocks are clearly visible and this is a key for our indirect search for the pipes.
Q: As the observed identification of the morphology of the carbonaceous layer and structure of the overlying sediments (or sills) is key, it would be helpful to have the authors give more support for their confidence in these impressive results. For the second case history, it would be helpful to show also the pseudo sections of the observed profiles. The goal would be to help the reader gain confidence in the resolving power of the method.
A: The supplementary pseudo section is not so useful in this case. Our aim is to show the main results of this study.
Reviewer 4 Report
Review of manuscript“Application of the Controlled Source Radio magnetotellurics(CSRMT) in the Study of Rocks Overlying Kimberlite Pipes in Yakutia/Siberia”
Тhe task of searching for kimberlite pipes in covered areas of the Yakutia kimberlite province is very difficult due to the significant heterogeneity of rocks overlying kimberlite pipes. This paper aims at the overlying rocks and the search for indirect indications of the presence of pipes by using the CSRMT data and their modeling. The paper is fairly well organized, But this paper is leak of innovation . Several issues must be addressed:
- Lack of keyand latestreferenceï¼›
- Supplementary forward modeling and inversion method of CSRMT
- Page 11 line 316 mentioned “Based on the results of 2D data inversion and modelling”, But I can’t find the 2D modelling part in the manuscript, Please supplementary the 2D modelling results.
Author Response
Dear reviewer,
Thank you for your comments and very useful suggestions for the improvement of the manuscript.
They were mainly taken into account and we made changes in figures and text.
Look please below our answers.
Yours sincerely,
Authors
Q: Тhe task of searching for kimberlite pipes in covered areas of the Yakutia kimberlite province is very difficult due to the significant heterogeneity of rocks overlying kimberlite pipes. This paper aims at the overlying rocks and the search for indirect indications of the presence of pipes by using the CSRMT data and their modeling. The paper is fairly well organized, But this paper is leak of innovation. Several issues must be addressed:
- Lack of key and latest reference
- Supplementary forward modeling and inversion method of CSRMT
- Page 11 line 316 mentioned “Based on the results of 2D data inversion and modelling”, But I can’t find the 2D modelling part in the manuscript, Please supplementary the 2D modelling results.
A: The key innovation of this publication is a new approach for indirect indications of the kimberlite pipes in this area and application of the CSRMT method to solve this complicated geological task. Of cause the CSRMT itself and 2D inversion is not innovation and corresponding references are there. Modelling was removed from the manuscript. One of the latest reference, recently published, is “Korobkov, I.G., Evstratov, A.A., Milshtein, E.D. Mafic volcanic structures of diamondiferous areas in the east side of Tunguska syneclise. Tomsk: SST, 2013, 270”. In this book the “windows in traps” is discussed from the geological point of view. Also, in the book “Kharkiv, A.D., Zinchuk, N.N., Kryuchkov, A.I. Geological and genetic basics of the mineralogical method of prospecting for diamond deposits. Moscow: Nedra, 1995, 348” such indicator of kimberlite pipes presence as the lowering of the top edge of carbonate hosting rocks is considered, also geological sides of this problem. So, we have demonstrated the possibility to map these indications using the new CSRMT method with the electrical dipole source.
Round 2
Reviewer 4 Report
This article is through the structures of rocks overlaying kimberlite pipes in two areas located in the Alakit Markha field of the Yakutia kimberlite province were studied by CSRMT field surveys. Based on the results of 2D data inversion the capabilities of the CSRMT method are demonstrated for the detailed study of rocks overlaying kimberlite pipes in two different areas of the Yakutia province. The main innovation is that the author proposes a method of indirectly searching for kimberlite by using "windows in traps" and depressions in the host rock, which are characteristics of kimberlite conduit positioning.
The language expression of this paper is relatively smooth, the chart is clear and meaningful, the conclusion is correct and innovative. However, there are still some minor errors in the text, which can be further improved:
- Line 42 "However, indirect indications of the presence of kimberlite pipes in the overlying and hosting rocks are often significant." Indirect indications are often significant. The text does not explain what Indirect indications are and should be supplemented.
- The representation of legend 2 in Figure 1 is incorrect, please correct it.
- In line 220 and line 223, the serial number of the reference is reversed, please correct it.
- The inversion code used in one-dimensional inversion comes from the reference cited in line 220. Please supplement the algorithm principle of the inversion code.
- There are too few references. I hope to add. In the introduction, I will add the geophysical work done by the predecessors and how effective is it. Supplementary references in the geological background, the research summary and research progress of the mining area.
Author Response
Dear reviewer,
Thank you for your comments and very useful suggestions for the improvement of the manuscript.
They were mainly taken into account and we made changes in figures and text.
Look please below our answers.
- Line 42 "However, indirect indications of the presence of kimberlite pipes in the overlying and hosting rocks are often significant." Indirect indications are often significant. The text does not explain what Indirect indications are and should be supplemented. Fixed
- The representation of legend 2 in Figure 1 is incorrect, please correct it. Corrected
- In line 220 and line 223, the serial number of the reference is reversed, please correct it. Corrected
- The inversion code used in one-dimensional inversion comes from the reference cited in line 220. Please supplement the algorithm principle of the inversion code. We add some details. The reference is there as well.
- There are too few references. I hope to add. In the introduction, I will add the geophysical work done by the predecessors and how effective is it. Supplementary references in the geological background, the research summary and research progress of the mining area. References are added.
Please find the revised manuscript in the attachment.
Rest regards,
Authors.
